# Insights into the Contribution of Oxidation-Reduction Pretreatment for Mn_0.2_Zr_0.8_O_2−δ_ Catalyst of CO Oxidation Reaction

**DOI:** 10.3390/ma16093508

**Published:** 2023-05-02

**Authors:** Denis D. Mishchenko, Zakhar S. Vinokurov, Tatyana N. Afonasenko, Andrey A. Saraev, Mikhail N. Simonov, Evgeny Yu. Gerasimov, Olga A. Bulavchenko

**Affiliations:** 1Boreskov Institute of Catalysis SB RAS, Lavrentiev Ave., 5, Novosibirsk 630090, Russia; q14999@yandex.ru (D.D.M.); smike@catalysis.ru (M.N.S.);; 2Synchrotron Radiation Facility SKIF, Boreskov Institute of Catalysis SB RAS, Nikol’skiy Prospekt 1, Kol’tsovo 630559, Russia; 3Center of New Chemical Technologies, Boreskov Institute of Catalysis SB RAS, Neftezavodskaya, 54, Omsk 644040, Russia

**Keywords:** Zr-Mn-solid solution, operando XRD, exsolution, Mn surface enrichment, zirconia, CO oxidation reaction

## Abstract

A Mn_0.2_Zr_0.8_O_2−δ_ mixed oxide catalyst was synthesized via the co-precipitation method and studied in a CO oxidation reaction after different redox pretreatments. The surface and structural properties of the catalyst were studied before and after the pretreatment using XRD, XANES, XPS, and TEM techniques. Operando XRD was used to monitor the changes in the crystal structure under pretreatment and reaction conditions. The catalytic properties were found to depend on the activation procedure: reducing the CO atmosphere at 400–600 °C and the reaction mixture (O_2_ excess) or oxidative O_2_ atmosphere at 250–400 °C. A maximum catalytic effect characterized by decreasing T_50_ from 193 to 171 °C was observed after a reduction at 400 °C and further oxidation in the CO/O_2_ reaction mixture was observed at 250 °C. Operando XRD showed a reversible reduction-oxidation of Mn cations in the volume of Mn_0.2_Zr_0.8_O_2−δ_ solid solution. XPS and TEM detected the segregation of manganese cations on the surface of the mixed oxide. TEM showed that Mn-rich regions have a structure of MnO_2_. The pretreatment caused the partial decomposition of the Mn_0.2_Zr_0.8_O_2−δ_ solid solution and the formation of surface Mn-rich areas that are active in catalytic CO oxidation. In this work it was shown that the introduction of oxidation-reduction pretreatment cycles leads to an increase in catalytic activity due to changes in the origin of active states.

## 1. Introduction

Manganese based catalysts are viable candidates to replace noble metal supported catalysts for oxidation reactions. Despite the high catalytic activity, noble metals are expensive, prone to deactivation during sintering [1,2], and are susceptible to sulfur and chlorine poisoning [3,4]. Manganese oxides show sufficient catalytic activity, are environmentally friendly, and have a significantly lower cost compared to noble metals [5,6]. Their field of application includes the removal of volatile organic compounds and CO from plant wastes and vehicles exhaust gases [6,7,8,9,10]. The high catalytic activity of Mn oxides is based on the ability of Mn to easily change the oxidation state from 2+ to 4+, which leads to a high lattice oxygen storage capacity in the oxide [11,12]. In addition, Mn oxide catalysts supported on Al_2_O_3_, SiO_2_, CeO_2_, TiO_2_, (Ce,Zr)O_2_ have been widely used in oxidation reactions [13,14,15,16,17,18,19,20,21,22,23]. This support affects the structural, microstructural, and redox properties of an Mn oxide catalyst due to the stabilization of Mn oxide nanoparticles on the surface of a support and the partial interaction with an active component [20,24]. Manganese-based mixed oxides exhibit improved catalytic properties in oxidation reactions compared to single-component catalysts. According to the reports of Lopez et al. [25], the high activity of MnOx-ZrO_2_ is due to the fact that ZrO_2_ stabilizes manganese in the catalytically active Mn^4+^ state, while the presence of manganese promotes the formation of a metastable tetragonal modification of ZrO_2_ with a high specific surface area. As it is known, there are three ZrO_2_ polymorphs with monoclinic, tetragonal, and cubic structures. The tetragonal modification of ZrO_2_ provides a higher activity of the supported component in various catalytic reactions [26,27,28]. Mn doping of ZrO_2_ leads to a creation of oxygen vacancies in the resulting solid solution (Mn_x_Zr_1−x_O_2−δ_), improving the overall lattice oxygen mobility [25,29,30,31,32]. Furthermore, there is a possibility of dispersed active MnO_x_ formation on the surface of the binary oxide [33]. There are two types of active phases for these catalysts: (1) the Mn_x_Zr_1−x_O_2−δ_ solid solution, (2) MnO_x_ on the surface of the solid solution. Depending on the preparation conditions, one of the active states dominates, or several types of active states coexist. At a relatively low Mn content (8 wt%), catalysts prepared via the impregnation method and containing MnO_x_ species exhibit a better catalytic performance than catalysts based on Mn_x_Zr_1−x_O_2−δ_ synthesized via co-precipitation [18]. With the increase in the Mn content, the co-precipitated catalysts predominantly show an improved catalytic performance [30,34]. Catalysts with the molar composition Mn:Zr = 4:6–6:4 exhibited the maximum catalytic activity in oxidation reactions [32,33,34,35]. With an increase in the total amount of Mn in the catalyst, the number of low valence (2+, 3+) manganese ions introduced into zirconia grows, promoting the generation of oxygen vacancies and increasing the overall activity in oxidation reactions [31]. The introduction of Mn into zirconia can also lead to a decrease in the size of crystallites and an increase in the specific surface area, therefore increasing catalytic activity [36]. For catalysts with high Mn loading, along with a solid solution, crystalline manganese oxides such as Mn_2_O_3_ and Mn_3_O_4_ are observed [30,33,34,35]. However, it is difficult to detect the formation of highly dispersed MnO_x_ and unequivocally attribute catalytic activity to one or another active component (dispersed MnO_x_ oxides or the Mn_x_Zr_1−x_O_2−δ_ solid solution). To eliminate the interfering influence of the presence of different phases, a single-phase mixed oxide containing a smaller amount of manganese Mn_0.2_Zr_0.8_O_2−δ_ was utilized to determine the correlations between activity and the nature of the active phases. The Mn_0.2_Zr_0.8_O_2−δ_ solid solution was treated in air, resulting in the partial decomposition of the initial oxide and exsolution of Mn ions from the lattice with the formation of highly dispersed MnO_x_ on the surface of the oxide support [37]. Such an approach to catalyst preparation based on the decomposition of a solid solution has been developed recently [38]. Typically in this method, the supported nanoparticles are formed from the solid solution of the host oxide via a redox exsolution [39]. The resulting nanoparticles are more uniformly distributed over the support surface and have a narrower size distribution compared to conventional supported catalyst synthesis methods, such as vapor deposition or impregnation. Usually, the driving force for the exsolution process is a change in oxidation state and, correspondingly, the ionic radius of guest elements. In the case of Mn_x_Zr_1−x_O_2−δ_, the driving force of catalyst decomposition during calcination in air is the change in the oxidation state of the Mn cation in the volume of mixed oxide [37]. Pretreatment in a reductive atmosphere accelerates the reduction of the Mn cation to Mn^2+^ and the migration of Mn cations to the surface of the mixed oxide [36].

Herein, we focus on the preparation of an active catalyst via the decomposition of a single-phase Mn_0.2_Zr_0.8_O_2−δ_ solid solution during redox treatment. To understand how pretreatment conditions affect catalytic properties, the oxidation environment (O_2_/reaction mixture) was varied. The catalytic properties were tested in a CO oxidation reaction. In order to obtain detailed information about the structural evolution of oxides and its effect on catalytic activity in the reaction of CO oxidation, we used operando X-ray diffraction (XRD) with mass spectrometry (MS) for the gas phase monitoring. The physicochemical properties of the catalyst before and after the pretreatment were studied by high-resolution transmission electron microscopy (HRTEM), X-ray absorption near edge structure (XANES), and X-ray photoelectron spectroscopy (XPS).

## 2. Materials and Methods

### 2.1. Synthesis

The Mn0.2Zr0.8 catalyst was prepared by a co-precipitation method. Precipitation was carried out with the vigorous stirring of a mixed Mn(NO_3_)_2_ ZrO(NO_3_)_2_ solution by the dropwise addition of a NH_4_OH solution until pH 10 was reached. The resulting precipitate was filtered off and washed with water on a filter to pH 6–7. The sample was dried at 120 °C for 2 h and then calcined in a muffle furnace at 600 °C for 4 h. The Mn/Zr molar ratio was 1:4.

### 2.2. Catalytic Tests

Catalytic tests were carried out using a flow circulation setup (BI-CATr, Boreskov Institute of Catalysis, Novosibirsk, Russia) complying with the principles of the non-gradient recycling reactor, which allows for measurement of the catalytic activity with a high accuracy due to the absence of both heat and concentration gradients inside the catalyst bed [40]. A detailed description of this reactor can be found elsewhere [41]. The setup was equipped with a circulation pump (capacity of 1000 L per hour) and a fluidized sand bed thermostat. The inlet gas mixture was preliminarily heated up to the temperature of the reactor, while the outlet mixture was cooled down to the circulation pump temperature. Temperature control was achieved using an Rh/Pt-type thermocouple located in the outlet reactor zone. The reactor was charged with 0.445 g of the catalyst mixed with quartz to a total volume of 1 mL. Helium was used as the carrier gas. All gases (CO, O_2_, He) were cleaned before analysis using a BI-GAS cleaner setup (Boreskov Institute of Catalysis, Russia) [42]. The analysis of the reaction products’ concentration in the outlet reaction mixture was performed using a Test-1 gas analyzer (Boner, Russia) equipped with IR and electrochemical sensors.

The catalyst was studied in the CO oxidation reaction both freshly prepared and after various pretreatments; the initial reaction mixture was 1% CO + 2% O_2_ in He. In the course of the reaction, the oxygen concentration was 4 times in excess of the stoichiometric to ensure the oxidizing conditions.

In step 1 heating to 270 °C was carried out followed by cooling in the reaction mixture.

In step 2 the catalyst was first pretreated in a reducing atmosphere (10% CO in He, total flow rate 100 mL/min) while raising the temperature from room temperature to 400 °C at a rate of 10 °C/min and holding at 400 °C for 1 h, after which the catalyst was cooled to room temperature and the catalytic activity was measured similarly to step 1.

In step 3 the catalyst was pre-treated in a reducing atmosphere (10% CO in He, total flow rate 100 mL/min) while raising the temperature from room temperature to 400 °C at a rate of 10 °C/min and holding at 400 °C for 1 h, after which oxidation was carried out in 10% O_2_ in He; the total flow rate was 100 mL/min, with holding at 400 °C for 1 h, after which the catalyst was cooled to room temperature and the catalytic activity was measured similarly to step 1.

In step 4 the catalyst was first pretreated in a reducing atmosphere (10% CO in He, total flow rate 100 mL/min) while raising the temperature from room temperature to 600 °C at a rate of 10 °C/min and holding at 600 °C for 1 h, after which the catalyst was cooled to room temperature and the catalytic activity was measured similarly to step 1.

### 2.3. Operando XRD

The operando XRD study was performed using a Bruker D8 Advance diffractometer (Bruker, Karlshruhe, Germany) equipped with a Lynxeye linear detector. The diffraction patterns were obtained in a θ/θ geometry using Ni-filtered CuK_α_ radiation (λ = 0.15418 nm). Diffraction patterns during stepwise heating/cooling (at a rate of 12 °C/min) were collected in a 2θ range of 21–66° with a step of 0.05° and a counting time of 7 s at each point using a XRK900 reactor chamber (Anton Paar, Graz, Austria). Additionally, the sample was scanned at ambient conditions after the operando experiment in a 2θ range of 25–90° with a step of 0.03° and a counting time of 12 s at each point. The phase composition, unit cell parameters, and coherent scattering region (CSR) of the sample were calculated using GSAS-II software packages (v. 5500) [43]. The phase analysis was performed using the ICDD PDF-2 database [44].The gas flow during the measurements was monitored using a UGA-100 quadrupole-type gas analyzer (Stanford Research System, Sunvyvale, California, USA) connected to the reactor outlet through a PEEK capillary. 

The operando XRD study was conducted in several consequent steps (sample measured at RT in steps 1, 18). In steps 2–4, 6–8, 11–13, and 15–17 the catalyst was studied in the CO oxidation reaction similarly to the catalytic tests at temperatures of 150, 200 and 250 °C. Steps 5, 9 + 10, and 14 repeated the catalytic tests from steps 2, 3, and 4, correspondingly.

### 2.4. XPS

The XPS measurements were performed on a photoelectron spectrometer (SPECS Surface Nano Analysis GmbH, Berlin, Germany) equipped with a PHOIBOS-150 hemispherical electron energy analyzer and an XR-50 X-ray source with a double Al/Mg anode. The core-level spectra were obtained using AlKα radiation (hν = 1486.6 eV) under ultrahigh vacuum conditions. The binding energy (E_b_) of photoemission peaks was corrected to the Ce3d_3/2_-U’’’ peak (E_b_ = 916.7 eV) of cerium oxide. The curve fitting was performed using CasaXPS software (v. 2.3.25). The line shape used in the fit was the sum of the Lorentzian and Gaussian functions. A Shirley-type background was subtracted from each spectrum [45]. Relative element concentrations were determined from the integral intensities of the core-level spectra using the theoretical photoionization cross-sections according to Scofield [46].

### 2.5. XANES

X-ray absorption near edge structure (XANES) spectra of the Mn K-edge (6539 eV) were recorded at the Structural Materials Science beamline at the Kurchatov Synchrotron Radiation Source (National Research Center “Kurchatov Institute”), Moscow, Russia. The spectra were obtained at the Mn K-edge using a channel-cut Si(111) monochromator. All measurements were performed in the transmission detection mode at room temperature. The ionization currents in the ion chambers were converted by the current amplifiers Keithley 6487. The obtained data were analyzed using the ATHENA software (v. 0.9.26) [47].

### 2.6. HRTEM

HRTEM images were obtained using a ThemisZ Thermo Fisher Scientific microscope (Thermo Fisher Scientific, Eindhoven, the Netherlands) with a resolution of 0.7 Å, respectively. Elemental maps were obtained using a SuperX Thermo Fisher Scientific energy dispersive spectrometer. The samples for research were fixed on standard copper grids using ultra-sonic dispersion of the catalysts in ethanol.

### 2.7. Surface Area 

The specific surface area was calculated using the Brunauer–Emmett–Teller (BET) method using nitrogen adsorption isotherms measured at liquid nitrogen temperatures on an automatic Micromeritics ASAP 2400 sorptometer (Norcross, GA, USA).

## 3. Results

### 3.1. Catalytic Activity

Figure 1 shows the evolution of CO conversion under heating and cooling cycles for the Mn0.2Zr0.8 catalyst in its as-synthesized and after-activation states. For the as-synthesized catalyst the light-off curve shifts to the lower temperatures after the reaction cycle with T_50_ equaling 197 and 193 °C, respectively. Reduction treatments shift both light-off curves (heating/cooling) to the lower temperatures. However, the oxidative treatment (step 3) does not show hysteresis in the heating and cooling cycle. T_50_ changes with treatment steps in the following order: 193–171–176–180 °C for the cooling runs, 197–180–176–186 °C for the heating runs. The lowest T_50_ of 171 °C was observed after the second step, and further steps only decreased the activity of the catalyst. From the given data, it can be observed that the initial reduction (step 2) was an important step for the catalyst activation. The temperature of the reduction should be considered carefully, since the higher it is, the higher the sintering effect will be. The increase of the reduction temperature to 600 °C (step 4) caused the light-off curve to shift towards higher temperatures and, accordingly, a change in T_50_ towards higher values. Another important step is the treatment of the catalyst in the reaction mixture or in an O2/He environment. In the case of steps 2 and 4, catalyst activation occurs under the catalytic reaction conditions during heating up to 250 °C, leading to a shift on the light-off curve into the low temperature region during cooling mode (T_50_ moves from 180 to 171°C and from 186 to 180 °C for the second and fourth steps, correspondingly). For the third step, the oxidation treatment at 400 °C takes place before the catalytic test and the CO conversion curves are matched for the heating and cooling runs, indicating that there is no additional activation under reaction conditions. The comparison of activation in the reaction mixture (step 2 cooling mode, oxidation in CO/O_2_ mixture up to 250 °C) and in the O_2_/He environment (step 3, 10%O_2_/He at 400 °C) shows that the oxidation that occurred in the reaction mixture and at a lower temperature up to 250 °C is better for catalyst activation than the direct oxidation at a higher temperature. As a result, the T_50_ value at cooling for the second step equal to 171 °C was lower than that of the third step equal to 176 °C (Figure 1).

To further elucidate the reasons for such catalytic behavior after different treatments, we conducted an operando XRD study under experimental conditions close to those of the catalytic tests.

### 3.2. Structural Operando XRD Characterisation

According to the XRD data, the catalyst showed no other phases during the operando experiments apart from the cubic fluorite (Mn, Zr)O_2_ based on the cubic ZrO_1.68–1.74_ (*a* = 5.128 Å, PDF #49-1642). Figure 2 shows the Rietveld refinement result for the initial sample and after the operando experiment. The obtained catalyst structure was used as a starting model to fit shorter scans (including only 111, 200, and 220 peaks) during the XRD operando measurements. The refinement of all patterns was performed using the position occupancies of Mn and Zr fixed to 0.2 and 0.8, correspondingly. The initial structure showed better refinement results with *a* = 5.0718(4) Å and CSR = 13 nm. The structure parameters after the operando experiments were *a* = 5.0716(3) Å and CSR = 15 nm. The decrease in the lattice parameter of oxide compared to the pure zirconia confirms the formation of an (Mn, Zr)O_2_ solid solution. Mn^3+^ cations (the ionic radius of 0.66 Å) substitute bigger Zr^4+^ cations (the ionic radius of 0.79 Å [48]) leading to a smaller lattice parameter. Based on the obtained parameter and literature data on the similar systems [25,36,49], the atomic content of Mn in a solid solution is ranged from 0.1 to 0.18. All the diffraction peaks showed a slight asymmetry towards lower diffraction angles, probably indicating some cation distribution, but this effect was not taken into account. The results of the operando XRD experiments are presented in Figure 3. The operando XRD study repeated the catalytic test procedure. At the beginning, the state of the catalyst was investigated under reaction conditions at 150, 200 and 250 °C (experimental steps 2, 3, and 4). The following experiments were conducted: the reductive pretreatment in CO at 400 °C (step 5)’ reaction at 150, 200, and 250 °C (steps 6, 7, and 8); reduction in CO at 400 °C, oxidation in O_2_/He (steps 9 and 10); reaction at 150, 200, and 250 °C (steps 11, 12, and 13); reduction in CO at 400 °C (step 14); reaction at 150, 200, and 250 °C (steps 15, 16, and 17). The XRD patterns (Figure 3a) show that the position of the 220 peak of fluorite depends on the temperature and gas environment. To compare the cell parameters (Figure 3b) correctly the thermal expansion contribution was subtracted from the fitted parameters. TEC for the Mn_0.2_Zr_0.8_O_2−δ_ was measured by heating the sample in He up to 600 °C and slowly cooling to room temperature and equated to 9.6 × 10^−6^ K^−1^ for the as-prepared sample. As can be seen from Figure 3b, the cell parameters depend on the environment (oxidative or reductive) and temperature. The cell parameter slowly decreases under reaction conditions (oxygen excess), indicating the oxidation of the Mn cations, and this effect is further enhanced at 400 °C in synthetic air. The opposite effect is observed for reductive media where the expansion of the fluorite cell occurs, indicating the reduction of Mn cations. Step 9 (reduction at 600 °C) shows no expansion compared to RT, probably due to two competitive processes—the reduction of Mn cations and lattice oxygen loss due to the high temperature.

Figure 3c shows the CO conversion of the catalyst after different pretreatments compared to the initial treatment. All the activation steps showed a positive effect on the catalytic activity in the reaction of CO oxidation. Adding the oxidation step 10 slightly increased CO conversion, while increasing the reduction temperature to 600 °C significantly decreased CO conversion at lower temperatures of 150 and 200 °C. Unfortunately, XRD, being a bulk method, cannot explain the mechanism of the catalyst activation since it, most likely, was caused by the surface changes. Specifically, the change in Mn content on the surface of the catalyst. Treating the catalyst in a reductive atmosphere depleted oxygen from the fluorite structure, causing not only a reduction in Mn ions but its migration to the surface due to a chemical potential change which could potentially led to an exsolution of MnO_x_ nanoparticles on the surface of the parent oxide. Further reoxidation could lead to a change in the surface state of the catalyst. In order to explore this, we employed surface-sensitive techniques, such as XPS and HRTEM, to study the initial catalyst and the catalyst after pretreatment in a reduction and oxidation atmosphere. XANES was also implemented due to the sensitivity to the valence states of Mn cations. The BET surface area for the initial catalyst was 58 m^2^/g; after pretreatment the surface area slightly decreased to 56 m^2^/g.

### 3.3. XPS and XANES Study

To study the chemical states and the relative concentrations of elements in the (sub)surface layers of the catalyst before and after pretreatment XPS was applied. Figure 4 shows the Mn*2p* core-level spectra of the catalysts. To identify the chemical state of manganese, the binding energy of the Mn*2p*_3/2_ peak and the intensity and position of satellites were used [50,51,52]; the asymmetry of the main peaks is due to the multielectron process. The fitting of the Mn*2p* spectra showed us, based on the Mn*2p*_3/2_ peak binding energy and intensity (as well as its satellites) [50,51,52], that manganese ions were present in 2+, 3+, and 4+ states, therefore, the Mn*2p* spectra were fitted by three Mn*2p*_3/2_–Mn*2p*_1/2_ doublets. The binding energies of the Mn*2p*_3/2_ peaks were equal to 641.0 eV, 641.7 eV, and 642.5 eV and assigned to Mn^2+^, Mn^3+^, Mn^4+^, respectively, which agrees well with the literature data [50,52,53,54,55,56,57,58,59,60,61,62]. The XPS analysis allowed us to estimate the fraction of manganese in different oxidation states and the relative surface concentrations (atomic ratios) of elements (Table 1). After the pretreatment, the content of Mn^2+^ and Mn^4+^ on the surface decreased, and the content of Mn^4+^ increased. XPS data showed that the ratio of [Mn]/[Zr] at the surface of the as-synthesized sample was 0.27; after the pretreatment the ratio changed to 0.3. For the solid solution with a stoichiometry of Mn_0.2_Zr_0.8_O_2−δ_ the ratio of Mn/Zr was 0.25. The surface concentration of Mn atoms exceeded the stoichiometric value by 8 and 20% for the fresh and treated catalyst, respectively. For Mn_x_Zr_1−x_O_2−δ_ oxide the surface was typically enriched with manganese cations, probably due to the unusual coordination of the Mn cation in the fluorite structure [37,63]. Along with the surface segregation of Mn cations, the average oxidation state of the Mn cations slightly increased from 2.77 to 2.84 after the pretreatment, due to the decrease in Mn^2+^ and the increase in Mn^4+^ cation fractions.

Figure 5 shows Mn K-edge XANES spectra of the catalyst before and after activation and the reference compounds for comparison. The Mn K-edge XANES spectra of the sample contained two regions: the main feature originating from the 1s→4p dipole transition; and the pre-edge region caused by 1s→3d transitions [64]. The main peak for the sample in both states was located at 6549.0 eV. Interestingly, the main peak had a doublet structure which was probably due to the superposition of absorption spectra from Mn located in different local environments. It is known that upon an increase in the oxidation state of Mn ions, the absorption edge shifts to higher energies almost linearly [12,64,65,66,67,68]. This fact allowed us to use a linear combination of the reference spectra to determine the oxidation states of Mn ions in the studied catalysts. It was revealed that the catalyst mainly comprised Mn^3+^ ions (70–75%) with the Mn^2+^ and Mn^4+^ content being about equal, and the activation steps did not change this amount, which agrees well with the XRD data. According to the XANES data, the average bulk oxidation state of Mn is around 3.0. This bulk oxidation state proved to be slightly different from the surface value (Table 1).

To sum up, the XRD and XANES methods do not demonstrate significant changes in the bulk structure. However, the XPS data results in a surface enrichment with the Mn cation and an increase in the average oxidation state of Mn cations. For further information about the surface morphology and cation content, we conducted a TEM study.

### 3.4. Morphology and Cation Surface Distribution

TEM images of the as-synthesized catalyst are presented in Figure 6. The as-synthesized catalyst consisted of round-shaped particles with sizes of about 20–40 nm joined into big aggregates up to 500 nm in size (Figure 6a,b). The separate particles consisted of differently orientated domains 5–10 nm in size (Figure 6b,c). Figure 6b shows that the domains contain an interplanar distance of 2.99 Å corresponding to the (111) plane of a solid Mn-Zr solution with a fluorite structure. EDS mapping showed a predominantly uniform distribution of Zr and Mn over the sample volume (Figure 6d).

Figure 7 shows TEM images of the catalyst after the pretreatment. After the catalytic test, the morphology of the catalyst particles stayed the same (Figure 7a,b). The selected particle contains interplanar distances of 2.96 and 2.54 Å, which are attributed to (111) and (200) planes of the Mn-Zr solid solution with a fluorite structure (Figure 7b). However, in the surface layers the interplanar distance (d_hkl_) dramatically changes from 2.96 to 3.36 Å (Figure 7b). The appearance of d = 3.36 Å can be attributed to the (201) plane of MnO_2_ (PDF # 42-1316). It is worth noting that EDX mapping detected enrichment of the surface by Mn atoms (Figure 7c,d). The Mn signal was not distributed over the sample homogeneously (Figure 7d), indicating the partial segregation of Mn and Zr atoms. Within the volume of one particle, the HAADF STEM images (Figure 7c) illustrate areas containing an interplanar distance of 3.36 Å with a size of 1–2 nm, showing the formation of surface domains with an MnO_2_ structure. However, XRD did not detect reflections of the MnO_2_. Interestingly, a similar effect was observed for the Mn-Ce oxide catalyst. TEM showed the formation of highly disperse MnO_2_ areas after calcination for a long time in air [69].

## 4. Discussion

In this work, the main idea was to prove that a reduction pretreatment leads to the partial decomposition of an Mn_0.2_Zr_0.8_O_2−δ_ solid solution with enrichment of the surface with Mn, while further reoxidation was believed to cause the formation of active in CO oxidation reaction MnO_x_ oxides. Catalytic tests did show that the use of reduction-oxidation pretreatments improved catalytic properties in the reaction of CO oxidation (Figure 1). The first reduction step was an important stage for the catalyst activation. Depending on the reduction temperature the degree of reduction changed. An increase in the reduction temperature from 400 to 600 °C led to an increase in T_50_ from 171 to 180 °C. Simultaneously, operando XRD data showed that under a reductive CO environment the lattice parameter increased. A change in the lattice parameter corresponds to a change in the valence state of Mn cations in the Mn_0.2_Zr_0.8_O_2−δ_ structure, lattice oxygen loss, or the exit of Mn ions from the volume of solid solution [36]. The second oxidation step occurred directly under reaction conditions, since the reaction mixture contained an excess of oxygen (CO:O_2_ = 1:2), or in an oxidative environment before the catalytic test (O_2_/He). Figure 1 illustrates that a reduction (2 and 4 steps) pretreatment alone is not sufficient for the improvement of the catalytic properties, because the light-off curves during cooling moved towards the low temperature. In the case of the third step in the catalytic tests, the light-off curves during cooling and heating were the same because the oxidative pretreatment took place before the catalytic reaction. After the oxidation pretreatment, the lattice parameter decreased, indicating oxidation of the Mn cations (Figure 2b). According to the XANES data before and after pretreatment, the bulk average oxidation state of Mn cations did not change (Table 1). However, XPS showed an increase in the average oxidation state of Mn cations on the surface from 2.77 to 3.84, simultaneously the content of Mn^2+^ decreased and Mn^4+^ increased (Table 1). In addition, TEM showed the appearance of some inhomogeneity in cation distribution and detected the formation of MnO_2_ oxide on the surface of Mn_x_Zr_1−x_O_2−δ_.

In the literature, there are restricted number of reports devoted to an exsolution of Mn cations from the volume of mixed oxides. For example, a cubic Nd_0.5_Ba_0.5_MnO_3_ was prepared aiming to exsolve Mn-based particles for a solid oxide fuel cells anode. It was shown that the perovskite undergoes a phase transition when the nanoparticles are released and forms a layered perovskite structure. The newly formed MnO nanoparticle-decorated perovskite was employed in the anode and exhibited an excellent power generation performance, even without any additional oxidation catalysts. [70] For a series of La_1−x_Ca_x_MnO_3_ perovskite, an exsolution of Mn_3_O_4_ from mixed oxide was observed after a methane oxidation reaction [71]. Gerasimov et al. simulated the La_0.2_Ca_0.8_MnO_3_ catalyst behavior and showed that at 1100 °C in a vacuum a partial decomposition of the La_1−x_Ca_x_ MnO_3_ oxide occurs. As a result, Mn_3_O_4_ nanoparticles appear, which are coherently bound with the perovskite phase with a defective structure [72]. In the case of La_1−x_Ca_x_MnO_3_ perovskites, the driving force of Mn cation migration towards the surface is the reduction of Mn^4+^ to Mn^3+^/Mn^2+^. In the case of an Mn_x_Zr_1−x_O_2_ based catalyst, the partial decomposition of a solid solution occurs due to a reduction of Mn ions in the volume of the fluorite structure. In Kroger-Vink notation, the formation of the point defect is ascribed by the following scheme: ZrO_2_ + Mn^2+^→ Mn_Zr_’’ + V_O_^●●^ after reduction vs. ZrO_2_ + Mn^3+^→ Mn_Zr_’ + 0.5V_O_^●●^ in the initial state, indicating an increase in the concentration of oxygen vacancies after a reduction of Mn cations. An increase in the cation radius of Mn cations and the formation of oxygen vacancies probably lead to the destabilization of the oxide fluorite structure and the exsolution of Mn ions. Further oxidation results in the aggregation of Mn cations and the formation of catalytically active and highly dispersed areas of MnO_2_ oxide on the surface of the parent oxide.

## 5. Conclusions

The Mn_0.2_Zr_0.8_O_2−δ_ based catalyst was synthesized via the co-precipitation method and the effect of pretreatment in an oxidative and reductive environment on its catalytic activity was tested in a CO oxidation reaction. To determine the activation conditions, the temperature of the reduction and oxidation and the treatment sequence were varied. As a result, the optimal conditions included a reduction temperature of 400 °C with further reoxidation in the reaction condition (oxygen excess). Operando XRD shows a strong dependence of the cell parameter of the solid solution on the temperature and gas environment, indicating changes in the Mn valence state and oxygen content. XPS and TEM methods detect the segregation of manganese cations on the surface of the solid solution, an increase in the surface concentration of Mn^4+^, and the formation of domains with an MnO_2_ structure. The utilized approach to the activation of the catalyst resulted in the enrichment of the surface of the mixed oxide with Mn cations with the further formation of manganese oxide nanoparticles.

## Figures and Tables

**Figure 1 materials-16-03508-f001:**
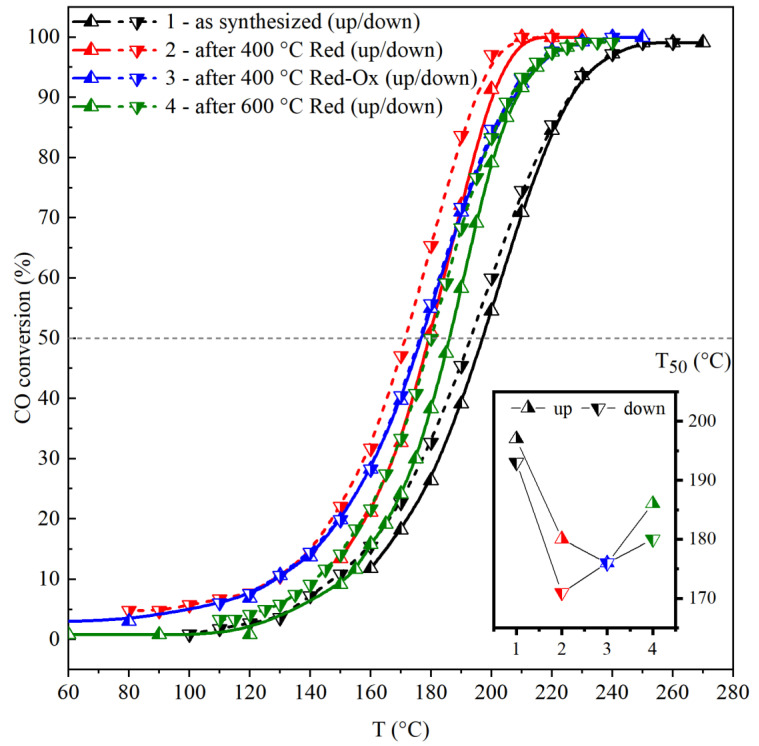
Light-off curves for the CO oxidation reaction over the Mn0.2Zr0.8 catalyst after different treatments. T_50_ are shown on the inset graph.

**Figure 2 materials-16-03508-f002:**
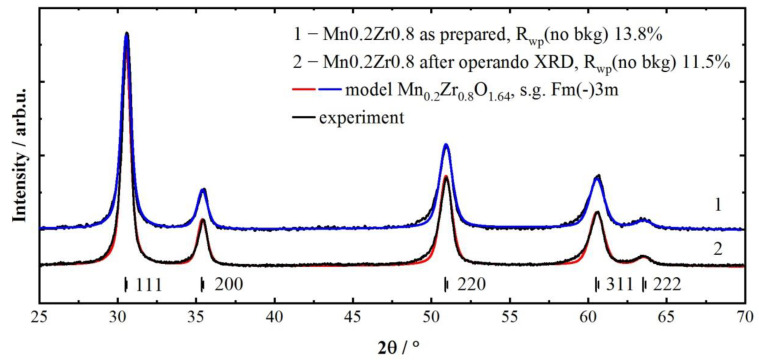
Rietveld refinement profile of the sample at RT after the operando experiment.

**Figure 3 materials-16-03508-f003:**
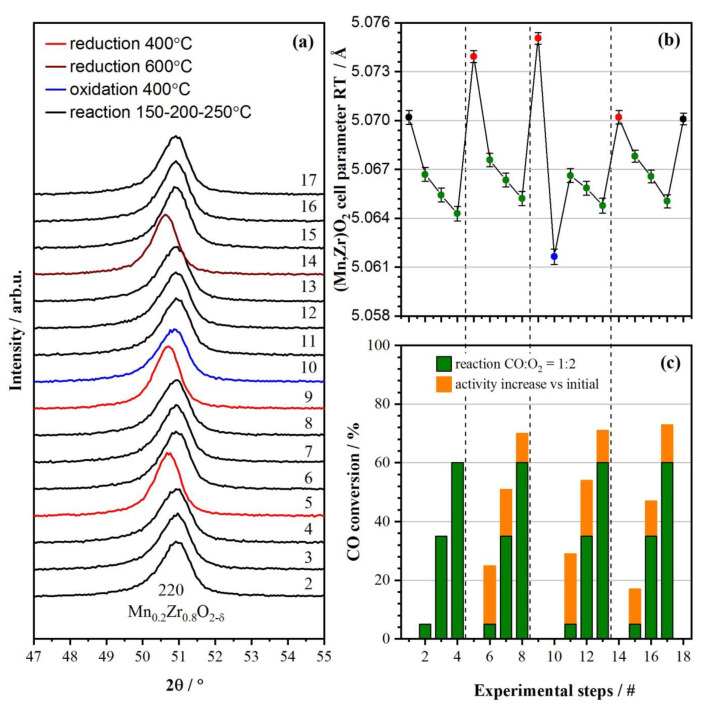
The operando XRD patterns (**a**). Unit cell parameter for the fluorite structure (**b**): black data points for the sample in the as-synthesized and treated states; green data points is for CO oxidation reaction at 150, 200, 250 °C; red data points is for the sample under reduction conditions; the blue data point is for the sample in oxidation conditions. CO conversion after consecutive pretreatments (**c**), #—number of a treatment step. Wavelength 0.15418 nm. Cell parameters are corrected to RT using the measured TEC of Mn_0.2_Zr_0.8_O_2-δ_ 9.6 × 10^−6^ K^−1^.

**Figure 4 materials-16-03508-f004:**
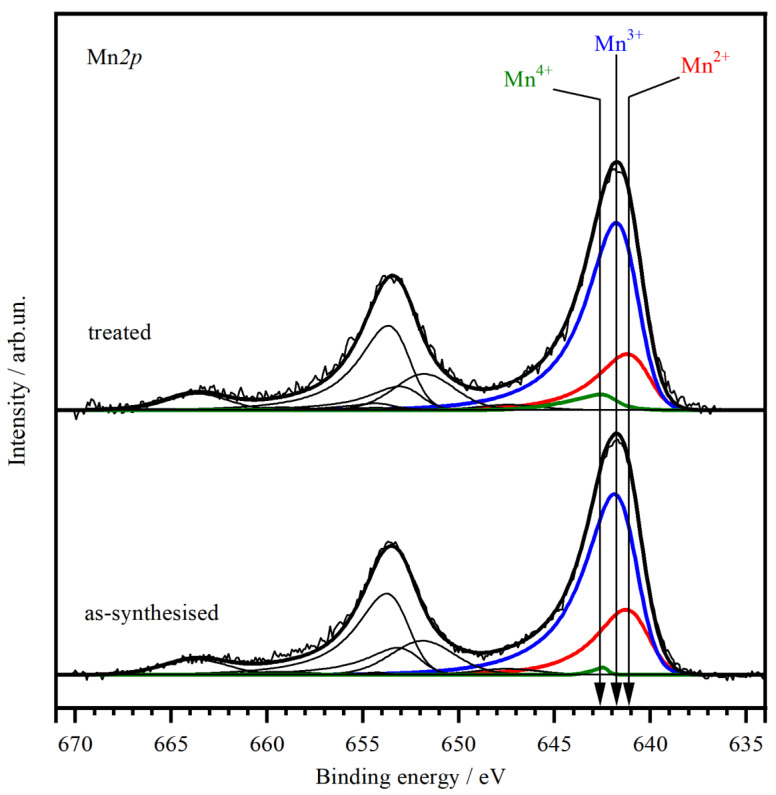
Mn2*p* core-level spectra of the catalyst in the as-synthesized state (**bottom**) and after the pretreatment (**top**).

**Figure 5 materials-16-03508-f005:**
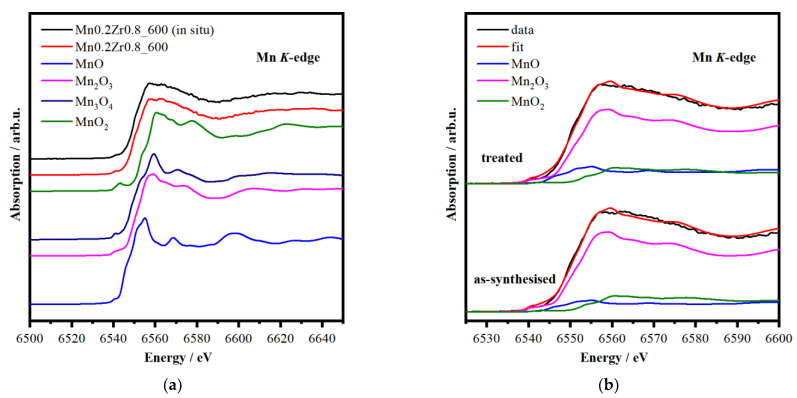
XANES Mn K-edge spectra of the catalyst before and after activation and the reference compounds (**a**). The results of the -fit for the as-synthesized sample ((**b**), bottom) and after pretreatment ((**b**), top).

**Figure 6 materials-16-03508-f006:**
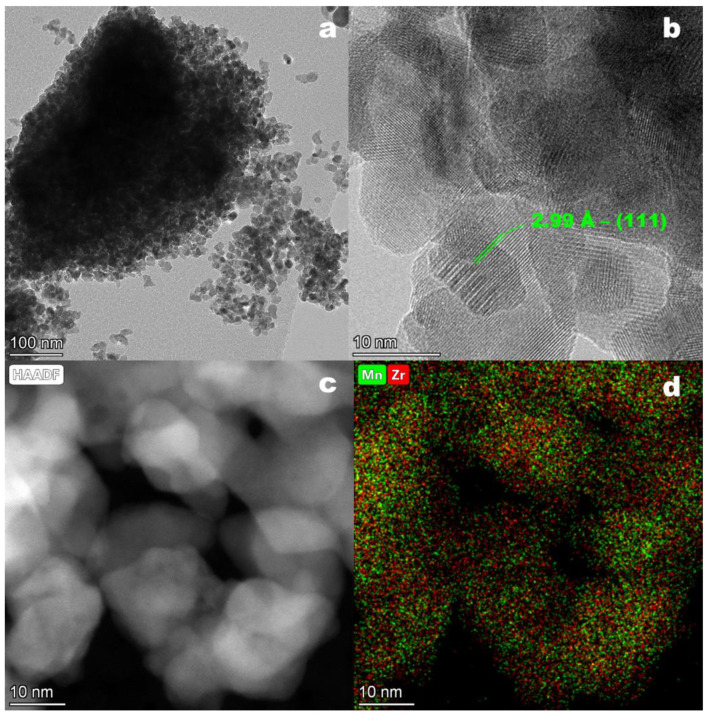
TEM (**a**,**b**), HAADF (**c**) images; EDS mapping pattern of Mn (green) and Zr (red) (**d**) for the as-synthesized catalyst.

**Figure 7 materials-16-03508-f007:**
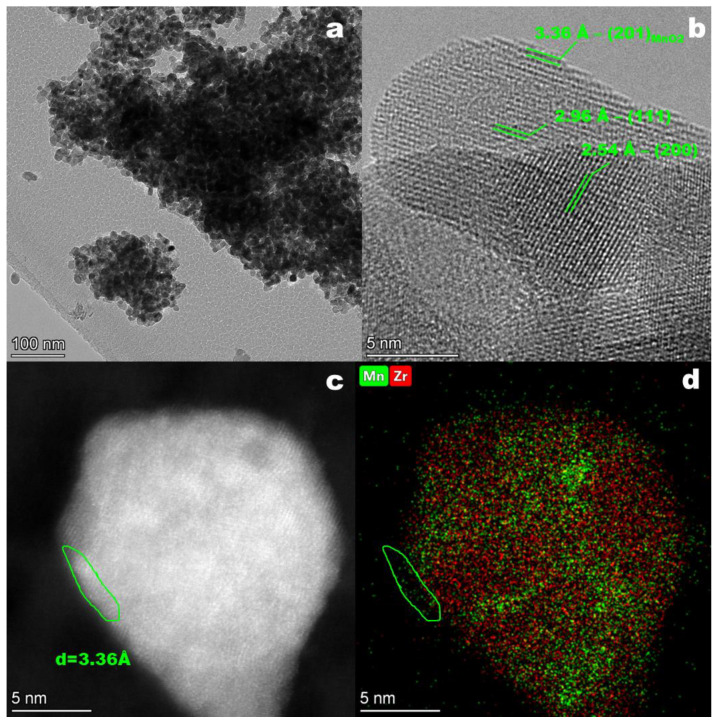
TEM (**a**,**b**), HAADF (**c**) images; EDS mapping pattern of Mn (green) and Zr (red) (**d**) for the treated catalyst.

**Table 1 materials-16-03508-t001:** Content of Mn and Zr atoms on the surface of the catalyst before and after pretreatment; average surface and bulk oxidation state of Mn cations from XPS and XANES results.

Sample	XPS				
	[Mn]/[Zr]	[Mn^2+^], %	[Mn^3+^], %	[Mn^4+^], %	Average Surface Oxidation State of Mn
As-synthesized	0.27	24	75	1	2.77
After the pretreatment	0.30	20	75	5	2.84

## Data Availability

The data presented in this study are available on request from the corresponding author.

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
