# Peer review of "Insights into the Contribution of Oxidation-Reduction Pretreatment for Mn0.2Zr0.8O2−δ Catalyst of CO Oxidation Reaction"

_materials, 2023, doi:10.3390/ma16093508_

Round 1
Reviewer 1 Report
This paper describes that pretreatment of Mn0.2Zr0.8O2-d by reductive atmosphere (CO in He) produced MnO2 species on the catalyst surface during CO oxidation, which contributed to the improvement of catalytic activity for CO oxidation. The Mn species on the catalyst surface was characterized by XRD, XPS, XAFS, and TEM. Although the difference in structure and Mn species between as-synthesized and pretreated under reductive conditions were disclosed, the reason for the inferior activity of re-oxidized and reduced at 600 degree C were not discussed based on the characterization results. In addition, the interpretation and/or explanation of the experimental results should be revised. Therefore, I think the paper can be accepted in principle, but the major revision is required. Please see the following comments.
1) In introduction, the authors stated that Mn-based oxides are environmentally friendly and have lower cost compared to noble metal. It is true, but the working temperatures of noble metals for CO oxidation are significantly lower than Mn-based oxides. Thus, noble metal catalysts for CO oxidation cannot be replaced by Mn-based oxides. Introduction should be rewritten.
2) In Figure 1, the catalytic activity at step 3 decreased the catalytic activity from step 2, and the authors concluded that re-oxidation at 400 degree C was detrimental effect. However, in Figure 3c, the CO conversions of #11-13 (after oxidation at 400 degree C) were slightly improved. The conversion curve of step 3 for heating up is better than that of step 2 for heating up. Therefore, the results of “detrimental effect of re-oxidation at 400 degree C” cannot be simply concluded. Explanation of the differences in the activity between heating up and cooling down with surface structure.
3) In p. 8, the authors stated that the content of Mn2+ and Mn4+ on the surface decreased and the content of Mn4+ increased after the pretreatment. Does Mn4+ decrease or increase? According to the experimental section, the “pretreatment” means reduction in CO/He flow. If it is so, the oxidation of Mn3+ species is just reduced whereas Mn is removed from the Mn0.2Zr0.8O2-d lattice. How Mn3+ was oxidized under reductive conditions? If the Mn species was re-oxidized during CO oxidation, the explanation should be revised. The characterization was performed just after treated in CO/He or after the pretreatment followed by CO oxidation?
4) In Figure 4, the spectra of treated and as-synthesized may be opposite. Mn4+ species of as-synthesized seems to be greater than the treated sample. In Table 1, The XPS peak ratio of Mn2+/Mn4+ for pretreated sample was 10/20. However, Mn2+ peak seems much greater than that of Mn4+ in Fig. 4 whatever for both spectra. In addition, the Mn3+ peak of the upper spectra seems to be more than 90%, but Mn3+ species was 66 or 70%. In any case, the peak fitting and quantitative analysis are not reliable.
5) Average surface oxidation state of Mn was estimated to be 2.84 and 3.10 for as-synthesized and pretreated samples by XPS. However, the XANES analysis revealed the opposite results. Even though there is a difference in surface and bulk, the oxidation state is supposed to not reverse. More convincing explanation is needed.
6) In Fig. 5b, it is hard to see the difference in spectra between as-synthesized and pretreated.
Author Response
Response to Review
We are thankful to the referee for useful comments. We made revision of the text according to the recommendations.
This paper describes that pretreatment of Mn0.2Zr0.8O2-d by reductive atmosphere (CO in He) produced MnO2 species on the catalyst surface during CO oxidation, which contributed to the improvement of catalytic activity for CO oxidation. The Mn species on the catalyst surface was characterized by XRD, XPS, XAFS, and TEM. Although the difference in structure and Mn species between as-synthesized and pretreated under reductive conditions were disclosed, the reason for the inferior activity of re-oxidized and reduced at 600 degree C were not discussed based on the characterization results. In addition, the interpretation and/or explanation of the experimental results should be revised. Therefore, I think the paper can be accepted in principle, but the major revision is required. Please see the following comments.
1) In introduction, the authors stated that Mn-based oxides are environmentally friendly and have lower cost compared to noble metal. It is true, but the working temperatures of noble metals for CO oxidation are significantly lower than Mn-based oxides. Thus, noble metal catalysts for CO oxidation cannot be replaced by Mn-based oxides. Introduction should be rewritten.
Answer: Indeed, catalysts based on noble metals in oxidation reactions are superior in activity to catalysts based on oxides of transition metals (including manganese). However, along with high catalytic activity, they have a number of disadvantages, such as high cost, tendency to caking [Nagai, Y.; Hirabayashi, T.; Dohmae, K.; Takagi, N.; Minami, T.; Shinjoh, H.; Matsumoto, S. Sintering Inhibition Mechanism of Platinum Supported on Ceria-Based Oxide and Pt-Oxide–Support Interaction, Simonsen, S.B.; Chorkendorff, I.; Dahl, S.; Skoglundh, M.; Sehested, J.; Helveg, S. Direct Observations of Oxygen-Induced Platinum Nanoparticle Ripening Studied by In Situ TEM. J. Am. Chem. Soc. 2010, 132, 7968–7975, doi:10.1021/ja910094r], and susceptibility to sulfur and chlorine poisoning [Cao, S.; Fei, X.; Wen, Y.; Sun, Z.; Wang, H.; Wu, Z. Bimodal Mesoporous TiO2 Supported Pt, Pd and Ru Catalysts and Their Catalytic Performance and Deactivation Mechanism for Catalytic Combustion of Dichloromethane (CH2Cl2). Applied Catalysis A: General 2018, 550, 20–27, doi:10.1016/j.apcata.2017.10.006., Lampert, J.; Kazi, M.; Farrauto, R. Palladium Catalyst Performance for Methane Emissions Abatement from Lean Burn Natural Gas Vehicles. Applied Catalysis B: Environmental 1997, 14, 211–223, doi:10.1016/S0926-3373(97)00024-6]. This limits the scope of their use and encourages the search for alternatives. Transition metal oxide catalysts are used as an inexpensive alternative to platinum and palladium for the purification of industrial emissions, vehicle exhaust gases from volatile organic compounds and CO [Li, W.B.; Wang, J.X.; Gong, H. Catalytic Combustion of VOCs on Non-Noble Metal Catalysts. Catalysis Today 2009, 148, 81–87, doi:10.1016/j.cattod.2009.03.007, Kapteijn, F.; Singoredjo, L.; Andreini, A.; Moulijn, J.A. Activity and selectivity of pure manganese oxides in the selective catalytic reduction of nitric oxide with ammonia. Applied Catalysis B, Environmental 1994, 3, 173-189, doi:10.1016/0926-3373(93)E0034-9.].
According to review recommendation, the introduction was rewritten. «Manganese based catalysts are viable candidates to replace noble metal supported catalysts for oxidation reactions. Despite the high catalytic activity, noble metals are expensive, prone to deactivation during sintering [1,2], and are susceptible to sulfur and chlorine poisoning [3,4]. Manganese oxides show sufficient catalytic activity, are environmentally friendly and have significantly lower cost compared to noble metals [5,6]. Their field of application includes the removal of volatile organic compounds and CO from plant wastes and vehicles exhaust gases [6–10]. High»
2) In Figure 1, the catalytic activity at step 3 decreased the catalytic activity from step 2, and the authors concluded that re-oxidation at 400 degree C was detrimental effect. However, in Figure 3c, the CO conversions of #11-13 (after oxidation at 400 degree C) were slightly improved. The conversion curve of step 3 for heating up is better than that of step 2 for heating up. Therefore, the results of “detrimental effect of re-oxidation at 400 degree C” cannot be simply concluded. Explanation of the differences in the activity between heating up and cooling down with surface structure.
Answer: We agree that “detrimental” is a bit strong term in this case. First of all it need to be clarified that Figure 3c illustrates conversion of #11-13 during heating run, and compared to the heating run for step 2 Fig.1 it shows better catalytic activity, but during cooling run it is vice versa. We actually wanted to say that oxidation occurred in reaction mixture and at lower temperature up to 250°C is better for catalyst activation than direct oxidation at 400°C. Text corrected accordingly. «Comparison of activation in the reaction mixture (step 2 cooling mode, oxidation in CO/O2 mixture up to 250 °C) and in O2/He environment (step 3, 10%O2/He at 400 °C) shows that the oxidation occurred in reaction mixture and at lower temperature up to 250°C is better for catalyst activation than direct oxidation at higher temperature.» Regarding «Explanation of the differences in the activity between heating up and cooling down with surface structure.» We do not have direct evidence of the change in morphology of the surface and only can speculate on the indirect data, that shows positive effect on the catalyst activity if the sample first reduced then oxidized in mild conditions.
3) In p. 8, the authors stated that the content of Mn2+ and Mn4+ on the surface decreased and the content of Mn4+ increased after the pretreatment. Does Mn4+ decrease or increase? According to the experimental section, the “pretreatment” means reduction in CO/He flow. If it is so, the oxidation of Mn3+ species is just reduced whereas Mn is removed from the Mn0.2Zr0.8O2-d lattice. How Mn3+ was oxidized under reductive conditions? If the Mn species was re-oxidized during CO oxidation, the explanation should be revised. The characterization was performed just after treated in CO/He or after the pretreatment followed by CO oxidation?
Answer: We absolutely agree with reviewer, in CO/He reduction of Mn cations should occur. In this work, first of all, we studied influence of different reduction and reoxidation pretreatments on the catalytic properties. Then we repeated the same pretreatments during operando XRD experiment. All pretreatments include reduction and oxidation step. Oxidation step could occur during oxidation in reaction media (CO+O2 at temperatures up to 250C) or additional oxidation in O2/He mixture. Operando XRD distinguished the reduction and oxidation steps. During reduction, change in the structure of mixed oxide is observed, a change in the lattice parameter corresponds to change in the valence state of Mn cations in the Mn0.2Zr0.8O2-δ structure, lattice oxygen loss or exit of Mn ions from the volume of sol-id solution. After oxidation steps, the structure of catalyst mostly returns to initial state. Unfortunately, from operando XRD, it is difficult to explain the mechanism of the catalyst activation since it, most likely, was caused by the surface changes. Therefore, catalyst after pretreatment, which includes reduction and oxidation steps, was investigated by XPS, XANES, and TEM.
To improve clarity of article, the description of the pretreatment has been clarified.
« Treating the catalyst in a reductive atmosphere depleted oxygen from fluorite structure causing not only reduction of Mn ions but its migration to the surface due to chemical po-tential change which could potentially led to exsolution of MnOx nanoparticles on the sur-face of the parent oxide. Further reoxidation could lead to change in surface state of catalyst. In order to explore this, we employed surface sensitive techniques such as XPS and HRTEM to study initial catalyst and after pretreatment in reduction and oxidation atmosphere. »
4) In Figure 4, the spectra of treated and as-synthesized may be opposite. Mn4+ species of as-synthesized seems to be greater than the treated sample. In Table 1, The XPS peak ratio of Mn2+/Mn4+ for pretreated sample was 10/20. However, Mn2+ peak seems much greater than that of Mn4+ in Fig. 4 whatever for both spectra. In addition, the Mn3+ peak of the upper spectra seems to be more than 90%, but Mn3+ species was 66 or 70%. In any case, the peak fitting and quantitative analysis are not reliable.
Answer: Indeed, the data at Figure 4 and Table 1 looks contradicted. We have checked and found that XPS fitting was incorrectly done. We have updated the fitting and Table 1.
5) Average surface oxidation state of Mn was estimated to be 2.84 and 3.10 for as-synthesized and pretreated samples by XPS. However, the XANES analysis revealed the opposite results. Even though there is a difference in surface and bulk, the oxidation state is supposed to not reverse. More convincing explanation is needed.
Answer: The average oxidation state was based on XPS and XANES spectra fitting and, of course, has the fitting error. For XANES data, the estimated values of average oxidation state are in the error range and should be evaluated as the same values. In case of XPS fitting, the increase of Mn4+ component is no doubt and should be taken in consideration.
6) In Fig. 5b, it is hard to see the difference in spectra between as-synthesized and pretreated.
Answer: We agree with the comment but now there is no opportunity to redone the measurements. The similarity of bulk structure of both samples leads to the similar XANES spectra and XRD as well.
Reviewer 2 Report
Review of the manuscript Materials-2347752for the Authors:
This article presents the study of mixed metal oxide catalysts (manganese and zirconia), prepared by an easy low-cost method. The experimental work is robust, the synthesis is novel just enough, and since the authors have presented their results quite nicely and concisely I can suggest this paper be accepted for publishing. Language and style are fine, no need for major interventions.
Title – Ok.
Abstract – Summarizes the paper and most important findings nicely.
Introduction – Offers a nice intro for the readers and highlights the importance and the core of the presented work.
Materials and methods – Quite detailed, offering all the necessary info.
Results and discussion – All of the results were summed up and explained in detail, and I must commend the figures as well, the authors stuck to the same/similar color schemes in different figures, which most people overlook but in my opinion is quite important.
Conclusions – Short but concise, as the conclusion should be.
Literature – Sufficient, up to date.
Author Response
We are thankful to the referee for comments.
Reviewer 3 Report
The present submission “Insights into the contribution of oxidation-reduction pretreatment for Mn0.2Zr0.8O2-δ catalyst of CO oxidation reaction” described the synthesis, pretreatment and application of Mn-Zr mixed oxide. In general, this manuscript is well written, and the characterizations are performed consecutively. On the other hand, it is significant and interesting to note that manganese can replace some expensive noble metals for preparation of oxidation catalyst. Lastly, the pretreatment carried out in this work shows important values in oxidation catalyst activation. However, I provide some comments for improvement of this work, and after revisions, I would like to recommend this work for publication in Materials.
At first, the review of the past literatures on manganese oxide and their use in oxidation of carbon monoxide is not enough. Manganese had more than one chemical valence, the Manganese ions were also incorporated into many supports including mesoporous silicate, zeolite, aluminum oxide, as well as carbon materials. These contents should be reviewed in Introduction.
Secondly, zirconium and its oxides should be reviewed in Introduction. Zirconium oxide may provide a solid support for catalyst durability, and the previous reports should be mentioned.
Next, in operando XRD, Fig. 2 and Fig. 3a, my personal opinion is that only zirconium oxide phase could be detected by XRD. How could authors determine the content of Mn by XRD including operando XRD? How could authors determine the formation of Mn-Zr mixed oxide phase? ICP and detailed XPS were recommended for further illustration. This part of characterization seemed very strange to readers.
Furthermore, in Fig. 4, the present peak splitting and fitting is artificial in association with software. However, how could authors prove the presence of Mn2+, Mn3+ and Mn4+? Raman spectroscopy is recommended for this point.
Lastly, the present pretreatment of synthesized catalyst would bring about the change of BET surface area. This important property would affect catalyst activity. Therefore, nitrogen physisorption is recommended to strengthen the results.
Overall, the characterizations were performed but not sufficient enough to give conclusions. Please kindly revise this work.
Thanks.
The manuscript is well written, just polish the language at the end of revision.
Author Response
Response to Review
We are thankful to the referee for useful comments. We made revision of the text according to the recommendations.
The present submission “Insights into the contribution of oxidation-reduction pretreatment for Mn0.2Zr0.8O2-δ catalyst of CO oxidation reaction” described the synthesis, pretreatment and application of Mn-Zr mixed oxide. In general, this manuscript is well written, and the characterizations are performed consecutively. On the other hand, it is significant and interesting to note that manganese can replace some expensive noble metals for preparation of oxidation catalyst. Lastly, the pretreatment carried out in this work shows important values in oxidation catalyst activation. However, I provide some comments for improvement of this work, and after revisions, I would like to recommend this work for publication in Materials.
At first, the review of the past literatures on manganese oxide and their use in oxidation of carbon monoxide is not enough. Manganese had more than one chemical valence, the Manganese ions were also incorporated into many supports including mesoporous silicate, zeolite, aluminum oxide, as well as carbon materials. These contents should be reviewed in Introduction.
Secondly, zirconium and its oxides should be reviewed in Introduction. Zirconium oxide may provide a solid support for catalyst durability, and the previous reports should be mentioned.
Answer: According to review recommendation, the introduction was rewritten. «Manganese based catalysts are viable candidates to replace noble metal supported catalysts for oxidation reactions. Despite the high catalytic activity, noble metals are expensive, prone to deactivation during sintering [1,2], and are susceptible to sulfur and chlorine poisoning [3,4]. Manganese oxides show sufficient catalytic activity, are environmentally friendly and have significantly lower cost compared to noble metals [5,6]. Their field of application includes the removal of volatile organic compounds and CO from plant wastes and vehicles exhaust gases [6–10]. High catalytic activity of Mn oxides is based on the ability of Mn to easily change the oxidation state from 2+ to 4+, which leads to a high lattice oxygen storage capacity in the oxide [11,12]. In addition, Mn oxide catalysts supported on Al2O3, SiO2, CeO2, TiO2, (Ce,Zr)O2 have been widely used in oxidation reactions [13–23]. Support affects the structural, microstructural, and redox properties of Mn oxide catalyst due to stabilisation of Mn oxide nanoparticles on the surface of support and partial interaction with an active component [20,24]. Manganese-based mixed oxides exhibit improved catalytic properties in oxidation reactions compared to single-component catalysts. According reports of Lopez et al. [25], the high activity of MnOx-ZrO2 is due to the fact that ZrO2 stabilizes manganese in the catalytically active Mn4+ state, while the presence of manganese promotes the formation of a metastable tetragonal modification of ZrO2 with a high specific surface area. As it is known, there are three ZrO2 polymorphs with monoclinic, tetragonal, and cubic structures. The tetragonal modification of ZrO2 provides a higher activity of the supported component in various catalytic reactions [26–28].»
Next, in operando XRD, Fig. 2 and Fig. 3a, my personal opinion is that only zirconium oxide phase could be detected by XRD. How could authors determine the content of Mn by XRD including operando XRD? How could authors determine the formation of Mn-Zr mixed oxide phase? ICP and detailed XPS were recommended for further illustration. This part of characterization seemed very strange to readers.
Answer: We apologize for a little misunderstanding there is indeed only one phase present, but it is a solid solution of (Mn,Zr)O2 with a fluorite structure that is based on the the cubic ZrO2. The most direct evidence of the formation of solid solution apart from the absence of MnOx phases is the fluorite lattice parameter that is significantly differs form pure zirconia due to the ion radius and valence state of Mn cations. Text corrected accordingly.
“The decrease in the lattice parameter of oxide compared to the pure zirconia confirms the formation of (Mn, Zr)O2 solid solution. Mn3+ cations (the ionic radius of 0.66 Å) substitute bigger Zr4+ cations (the ionic radius of 0.79 Å [31]) leading to a smaller lattice parameter. Based on the obtained parameter and literature data on the similar systems [9,18,32] the atomic content of Mn in solid solution is ranged from 0.1 to 0.18.”
Furthermore, in Fig. 4, the present peak splitting and fitting is artificial in association with software. However, how could authors prove the presence of Mn2+, Mn3+ and Mn4+? Raman spectroscopy is recommended for this point.
Answer: The presence of different Mn cations is proved by XPS and XANES spectroscopy as well. To fit the XPS spectra we have previously studied the reference Mn compound (MnO, Mn2O3, and MnO2). It should be noted that we have found the fitting mistake in XPS fitting and Figure 4 and Table 1 have been updated. Unfortunately, we have no opportunity to measure the Raman spectra of the samples. We thank for the advice and will try to use it in the next investigations.
Lastly, the present pretreatment of synthesized catalyst would bring about the change of BET surface area. This important property would affect catalyst activity. Therefore, nitrogen physisorption is recommended to strengthen the results.
Answer: According to reviewer recommendation, the change of BET surface area was investigated for as-prepared catalyst and after pretreatment. The specific surface area was calculated with the Brunauer–Emmett–Teller (BET) method using nitrogen adsorption isotherms measured at liquid nitrogen temperatures on an automatic Micromeritics ASAP 2400 sorptometer.Norcross, GA, USA). Initial Mn0.2Zr0.8 catalyst is characterized by the BET surface area of 58 m2/g, after pretreatment the surface area slightly decreases to 56 m2/g. This information has been added into the manuscript.
Round 2
Reviewer 1 Report
The revised manuscript can be accepted.
Reviewer 3 Report
After reading the revised version, I would like to recommend the present revised version for publication in Materials.